# Towards Safe Reinforcement Learning via Constraining Conditional Value at Risk

**Chengyang Ying** [1] **Xinning Zhou** [1] **Dong Yan** [1] **Jun Zhu** [1]

## Abstract

Though deep reinforcement learning (DRL) has obtained substantial success, it may encounter catastrophic failures due to the intrinsic uncertainty caused by stochastic policies and environment variability. To address this issue, we propose a novel reinforcement learning framework of CVaR-Proximal-Policy-Optimization (CPPO) by rating the conditional value-at-risk (CVaR) as an assessment for risk. We show that performance degradation under observation state disturbance and transition probability disturbance theoretically depends on the range of disturbance as well as the gap of value function between different states. Therefore, constraining the value function among states with CVaR can improve the robustness of the policy. Experimental results show that CPPO achieves higher cumulative reward and exhibits stronger robustness against observation state disturbance and transition probability disturbance in environment dynamics among a series of continuous control tasks in MuJoCo.

## 1. Introduction

Recently, reinforcement learning (RL) coupled with deep neural networks has achieved enormous success among a variety of tasks, ranging from playing Atari games (Mnih et al., 2016; 2013; 2015) and Go (Silver et al., 2016) to manipulating complex robotics in the real world (Kendall et al., 2019). However, those methods tend to exhibit considerable uncertainty and may result in catastrophic failures due to the great variability in both models and the environment (Heger, 1994; Coraluppi & Marcus, 1999). Several factors can be associated with this phenomenon, one is that traditional DRL

*Equal contribution [1]Dept. of Comp. Sci. & Tech., Institute for AI, BNRist Center, Tsinghua-Bosch Joint ML Center, Tsinghua University, Beijing, China. Correspondence to: Chengyang Ying <ycy17@mails.tsinghua.edu.cn>.

*Accepted by the ICML 2021 workshop on A Blessing in Disguise: The Prospects and Perils of Adversarial Machine Learning.* Copyright 2021 by the author(s).

only aims to maximize the cumulative reward without taking account of the variance of trajectories (Garcıa & Fernández, 2015), which may lead to serious consequences. This can be illustrated briefly in the case of self-driving, where the agent might try to achieve the highest reward by acting dangerously, e.g. driving along the edge of a curve. Also, the use of deep neural network to construct complicated mappings from high-dimension state space $\mathcal{S}$ to action space $\mathcal{A}$ in DRL algorithms can make them vulnerable to adversarial attacks, which has been reported by Huang (Huang et al., 2017).

To address the pessimistic policy problem and the inappropriate usage of variance in previous objective-modification methods (Garcıa & Fernández, 2015; Geibel & Wysotzki, 2005; Heger, 1994), we propose a new method combining RL algorithms with CVaR. While using the variance as penalty will eliminate both particularly good and bad trajectories, CVaR will only capture bad ones. Based on the integration of Proximal Policy Optimization (PPO) with CVaR, we construct our algorithm called CVaR-Proximal-Policy-Optimization(CPPO). We compare CPPO to multiple baselines among continuous control tasks in MuJoCo and provide theoretical analysis of its robustness under perturbations of observation and transition probability.

## 2. Background

In this section, we will briefly introduce Safe RL, CVaR and describe our motivation for adopting CVaR as a better metric of risk in Safe RL.

### 2.1. Safe RL

Given MDP $\mathcal{M} = (\mathcal{S}, \mathcal{A}, \mathcal{R}, P, \gamma)$, the goal of RL is to find the optimal policy $\pi_{\theta^*}$ with highest cumulative reward:

$$\max_{\theta} J(\pi_\theta) \triangleq E\left[D(\pi_\theta) \triangleq \sum_{t=1}^{\infty} \gamma^t R_t | \pi_\theta\right]. \quad (1)$$

However, eq. (1) only focuses on cumulative reward without taking account of the risk of the policy, which may cause catastrophic results. To address this problem, safe RL methods tend to make modifications to eq. (1) function in order to eliminate the uncertainty, which can be categorized

into the inherent uncertainty and the parameter uncertainty (García & Fernández, 2015).

The inherent uncertainty of RL refers to the transition dynamics in MDP, for example the agent might end up in completely different situations when repeating its action at the same state. Previous works such as (Heger, 1994; Gaskett, 2003) chose the worst-case criterion to deal with the inherent uncertainty:

$$\max_{\theta} J_{inh}(\pi_\theta) \triangleq \max_{\theta} \min_{\tau \sim \pi_\theta} \left[ D(\tau) \triangleq \sum_{t=1}^{\infty} \gamma^t R_t \right]. \quad (2)$$

The parameter uncertainty of RL denotes scenarios where the parameters of the MDP are unknown or there is a gap between the training and testing environments. Studies conducted by Nilim & El Ghaoui; Tamar et al. assume that the actual transition probability belongs to a set $\hat{\mathcal{P}}$ and consider the following equation:

$$\max_{\theta} \min_{\mathcal{P} \in \hat{\mathcal{P}}} J_{par}(\pi_\theta, \mathcal{P}) \triangleq E \left[ D(\pi_\theta) \triangleq \sum_{t=1}^{\infty} \gamma^t R_t | \pi_\theta \right]. \quad (3)$$

However, previous Safe RL methods suffer from serious drawbacks. On the one hand, focusing on the worst trajectories may cause over pessimistic behaviors. On the other hand, the direct usage of variance to penalize risk is another potential concern because it will not only eliminate the possibility of particularly bad trajectories, but also particularly good ones, and thus causes a drop in the agent's performance. Moreover, eq. (2) and eq. (3) are all *max-min problems*, which don't have general solutions and traditional methods usually require high computation complexity. With the incorporation of CVaR, we focus on increasing the agent's performance on relatively worse trajectories, which loosens the *max-min problem* to an constrained optimization problem.

## 2.2. CVaR

CVaR is a well-established metric for measuring risk in economy. It estimates the probability of the random variable to be an outlier with given threshold. First, we will clarify the definition of VaR and CVaR (Chow & Ghavamzadeh, 2014b):

**Definition 1 (VaR and CVaR)** *For a bounded-mean random variable Z, we can define its cumulative distribution function as F(z) = P(Z ≤ z). Then its Value at Risk (VaR) of confidence level $\alpha$ is:*

$$\mathrm{VaR}_\alpha(Z) = \min\{z | F(z) \geq \alpha\}, \quad (4)$$

*and its Condition Value at Risk(CVaR) of confidence level $\alpha$ is defined as the expectation of the $\alpha$-tail distribution of Z:*

$$\mathrm{CVaR}_\alpha(Z) = E\{z | z \geq \mathrm{VaR}_\alpha(z)\}. \quad (5)$$

Previous works have attempted to analyze the risk-MDP with CVaR. Chow and et al. propose gradient-based methods like policy gradient and actor critic to optimize loss of MDP as well as keeping the CVaR under certain value (Chow & Ghavamzadeh, 2014a; Chow et al., 2015a). They also propose methods based on value iteration and Bellman equation to deal with the optimization of risk-MDP with CVaR (Chow et al., 2015b). However, their work ignores the reward in MDP and thus can not be directly used in RL settings.

Compared with variance, CVaR is a better metric for estimating risk, because CVaR, by definition, can capture only the bad trajectories. Based on CVaR and PPO, we propose CPPO and achieve models with higher overall performance and stronger robustness.

## 3. Methodology

In this section, we'll formalize our objective as a constrained optimization problem and provide a gradient-based algorithm. Moreover, we'll provide theoretical analysis of our algorithm against adversarial noises.

### 3.1. Problem Formulation

In this paper, we define risk as the probability to generate low-reward trajectories.As RL focuses on maximizing expected cumulative reward while ignoring risk of the policy, we'd like to model our objective as maximizing the expected reward within a constrained region defined by CVaR.

However, the standard definition of CVaR portrays the expectation of the part with higher value in random variable $Z$, which is better to be low when $Z$ directly represents risk, such as risk MDP (Chow & Ghavamzadeh, 2014a; Chow et al., 2015a). But in general RL, the bad trajectories represent those with low cumulative reward. Thus the objective of our method is to maximize the total reward $J(\pi_\theta)$ as well as letting the expected reward of low-reward part of $D(\pi_\theta)$ be higher than a given lower bound. By the property of CVaR, we can use $-\mathrm{CVaR}_\alpha(-D(\pi_\theta))$ to represent the expected reward of the trajectories generated by $\pi_\theta$ with lower reward (see Appendix(A.1)).

As mentioned in Section 2.1, eq. (2) and (3) are intractable *max-min problems*. However, with the property of CVaR, we can equally transform eq. (2) as follow (see Appendix(A.1)):

$$\max_{\theta} J_{inh}(\pi_\theta) = \max_{\theta} \lim_{\alpha \to 1^-} \left[ -\mathrm{CVaR}_\alpha(-D(\pi_\theta)) \right]. \quad (6)$$

We can further loosen equation (6) by assigning $\alpha$ a fixed value, which reforms the original *max-min problem* into a solvable optimization problem. Furthermore, to address the pessimism in Safe RL, we balance between standard RL objective (eq. (1)) and Safe RL objective (eq. (6)) after

relaxation, which is the constrained optimization problem as below:

$$\max_{\theta} J(\pi_\theta)$$
$$s.t. - \text{CVaR}_\alpha(-D(\pi_\theta)) \geq \beta, \tag{7}$$

where $\alpha, \beta$ are hyper-parameters and we denote the best policy of this optimization problem (7) as $\pi_c(\alpha, \beta)$. We also reveal some properties of $\pi_c(\alpha, \beta)$ in Appendix(A.2).

### 3.2. Optimization

In this part, we'll use properties of CVaR to simplify our problem (7), which is a constrained optimization problem, to an unconstrained optimization problem.

First, we'll use properties of CVaR to simplify the raw problem (7) as the below theorem:

**Theorem 1** *We can deform the problem (7) above equivalently as:*

$$\min_{\theta, \nu} -J(\pi_\theta)$$
$$s.t. - \nu + \frac{1}{1-\alpha}E[(-D(\pi_\theta)+\nu)^+] \leq -\beta. \tag{8}$$

We provide the proof of Theorem 1 in Appendix(A.3). However, the problem (1) we now consider is still a constrained optimization problem. By using Lagrangian relaxation method (Bertsekas, 1999), we need to solve the saddle point of the function $L(\theta, \nu, \lambda)$:

$$\max_{\lambda \geq 0} \min_{\theta, \nu} L(\theta, \nu, \lambda) \triangleq -J(\pi_\theta)+$$
$$\lambda(-\nu + \frac{1}{1-\alpha}E[(-D(\pi_\theta)+\nu)^+] + \beta). \tag{9}$$

### 3.3. CVaR Proximal Policy Optimization

In this part, we will extend PPO (Schulman et al., 2017) from standard RL to our method with CVaR, named CVaR Proximal Policy Optimization(CPPO).

The key point of Policy Gradient method is to evaluate the gradient (Sutton et al., 1999), we use the methods in (Chow & Ghavamzadeh, 2014a) to compute the gradient of our objective function (9) and give the Theorem 2, of which the proof is in Appendix(A.4):

**Theorem 2** *The gradient of the objective function (9) with respected to $\nu, \theta, \lambda$ are as below:*

$$\partial_\nu L(\theta, \nu, \lambda) = -\lambda + \frac{\lambda}{1-\alpha}E_{\xi \sim \pi_\theta}\mathbf{I}\{\nu \geq D(\xi)\}) \tag{10}$$

$$\nabla_\theta L(\theta, \nu, \lambda)$$
$$= -E_{\xi \sim \pi_\theta}(\nabla_\theta \log P_\theta(\xi))\left(D(\xi) - \frac{\lambda}{1-\alpha}(-D(\xi)+\nu)^+\right) \tag{11}$$

$$\nabla_\lambda L(\theta, \nu, \lambda) = -\nu + \frac{1}{1-\alpha}E_{\xi \sim \pi_\theta}(-D(\xi)+\nu)^+ + \beta. \tag{12}$$

Based on PPO and the algorithm in (Chow & Ghavamzadeh, 2014a), we can use the gradient proven in Theorem 2 to propose the pseudo code of CPPO (see Appendix (B)):

### 3.4. Theoretical Analysis

In this part, we will analyze the robustness of policies against state observation noises as well as transition probability noises.

For every state $s \in \mathcal{M}$, we can define its discount future state distribution as $d_{\mathcal{M}}^\pi(s) = (1-\gamma)\sum_{t=0}^{\infty} \gamma^t P(s_t = s|\pi, \mathcal{M})$. First, we will consider the situation of state observation disturbance. Similar to the setting of SA-MDP (Zhang et al., 2020), we introduce adversary $\nu : \mathcal{S} \to \mathcal{S}$ to describe the disturbance of state and denote the policy disturbed by adversary $\nu$ as $\hat{\pi}_\nu$. We show that the difference of performance between $\pi$ and $\hat{\pi}_\nu$ can be calculated as below:

**Theorem 3** *For any policy $\pi$ and any adversary $\nu$, we have:*

$$J_{\mathcal{M}}(\pi) - J_{\mathcal{M}}(\hat{\pi}_\nu)$$
$$= \frac{\gamma}{1-\gamma}E_{s \sim d_{\mathcal{M}}^{\hat{\pi}_\nu}}E_{a \sim \pi(\cdot|\nu(s))}\left(1 - \frac{\pi(a|s)}{\pi(a|\nu(s))}\right)$$
$$E_{s' \sim P}V_{\mathcal{M},\pi}(s')$$
$$+ \frac{1}{1-\gamma}E_{s \sim d_{\mathcal{M}}^{\hat{\pi}_\nu}}E_{a \sim \pi(\cdot|\nu(s))}\left(1 - \frac{\pi(a|s)}{\pi(a|\nu(s))}\right)R(s,a). \tag{13}$$

*Furthermore, we can give an upper bound of it:*

$$|J_{\mathcal{M}}(\pi) - J_{\mathcal{M}}(\hat{\pi}_\nu)|$$
$$\leq \frac{\gamma}{1-\gamma}\max_s D_{TV}(\pi(\cdot|s), \pi(\cdot|\nu(s)))b$$
$$+ \frac{2}{1-\gamma}\max_s D_{TV}(\pi(\cdot|s), \pi(\cdot|\nu(s))\max_{s,a}|R(s,a)|. \tag{14}$$

*here $b = \max_s V_{\mathcal{M},\pi}(s) - \min_s V_{\mathcal{M},\pi}(s)$.*

The complete proof of Theorem 3 resembles the proof by (Kakade & Langford, 2002) in Appendix(A.5). We also prove the bound (14) is better than the bound in (Zhang et al., 2020) in Appendix(A.5).

Then we'll consider the situation of transition probability disturbance. Similar to Theorem 3, we can show that:

**Theorem 4** *For any policy $\pi$ and any disturbed environ-*

| Method \ Game | Ant-v3 | HalfCheetah-v3 | Walker2d-v3 | Swimmer-v3 | Hopper-v3 |
|---|---|---|---|---|---|
| VPG | 12.8± 0.0 | 896.9± 531.1 | 628.6± 229.4 | 48.3± 11.3 | 888.4± 209.5 |
| TRPO | 1625.4± 356.4 | 2073.8± 741.3 | 2005.6± 398.7 | 101.2± 29.3 | 2391.4± 455.3 |
| PPO | 3372.2± 301.4 | 3245.4± 947.3 | 2946.3± 944.3 | 122.0± 7.9 | 2726.0± 886.0 |
| CPPO(ours) | **3445.3± 325.6** | **3680.5± 1121.3** | **3079.2± 729.6** | **180.7± 48.5** | **3048.4± 134.1** |

*Table 1.* The cumulative reward (mean ± one std) of best policy trained by VPG, TRPO, PPO and CPPO in different MuJoCo games.

ment $\hat{\mathcal{M}} = (\mathcal{S}, \mathcal{A}, \hat{\mathcal{P}}, \mathcal{R})$, we have:

$$J_{\mathcal{M}}(\pi) - J_{\hat{\mathcal{M}}}(\pi)$$
$$= \frac{\gamma}{1-\gamma} E_{s \sim d_{\hat{\mathcal{M}}}^{\pi}} E_{a \sim \pi} E_{s' \sim \hat{P}} \left( 1 - \frac{P(s'|s,a)}{\hat{P}(s'|s,a)} \right) V_{\mathcal{M},\pi}(s'). \tag{15}$$

*Furthermore, we can give a upper bound of it:*

$$J_{\mathcal{M}}(\pi) - J_{\hat{\mathcal{M}}}(\pi)$$
$$\leq \frac{2\gamma}{1-\gamma} \max_{s,a} D_{TV}(P(\cdot|s,a), \hat{P}(\cdot|s,a))b, \tag{16}$$

*here* $b = (\max_s V_{\mathcal{M},\pi}(s) - \min_s V_{\mathcal{M},\pi}(s))$.

The complete proof of Theorem 4 in Appendix(A.5).

By theorem 3 and theorem 4, we can find out that the effects of state observation disturbance and transition probability disturbance on cumulative reward are both depend on the scale of disturbance as well as the gap of the value function between the best state and the worst state. Thus CVaR-based methods can control the value function of the worst state to improve the robustness of the policy.

## 4. Experiments

In this section, we will evaluate our method in a series of continuous control tasks in MuJoCo (Todorov et al., 2012). First, we introduce the environments we use and the benchmarks in sec. 4.1, as well as the evaluation metric in sec. 4.2. Then we present our experimental results in sec. 4.3.

### 4.1. Experiment Setup

**Tasks:** We choose MuJoCo as our experiments environment. As a robotic locomotion simulator, MuJoCo has an array of different continuous control tasks such as Ant, Walker2d, HalfCheetah, Hopper, Swimmer.

**Baselines:** We will compare our algorithm with the common on-policy algorithms like Vanilla Policy Gradient(VPG)(Sutton et al., 1999), Trust Region Policy Optimization(TRPO)(Schulman et al., 2015) and PPO(Schulman et al., 2017). And we use Adam (Kingma & Ba, 2015) to optimize all the parameters.

**Code:** We implement our algorithms and compare them with baseline based on SpinningUp (Achiam, 2018).

### 4.2. Evaluation Metric

**Evaluation:** First, we compare the cumulative reward of each algorithm in the training process, and their performance after convergence. For the trained models, in order to measure their robustness and safety, we compare their performance under transition probability disturbance and state disturbance respectively. For the transition probability perturbation, we notice that MuJoCo is a physical simulation engine, so we modify the mass of the agent to change the transition dynamics, and study the relationship between the the agent's performance and the mass of the agent. For state disturbance, we apply adversarial noise to the agent's observation to study the relationship between the agent's performance and the magnitude of the noise.

### 4.3. Experiments on MuJoCo

Since our policies are stochastic, we will train ten policies with different random seeds for each algorithm. First, we plot their mean and variance of the cumulative reward during training time (see Appendix (C.1)). We also record the best results of each algorithm in Table 1. For evaluating the robustness of trained policies, we plot the performance of agents under observation disturbance with varying noises size (see Appendix(C.2)). Finally, we appraise the robustness under transition probability disturbance (see Appendix (C.3)).

## 5. Conclusions

In this paper, we define risk as trajectories with low cumulative reward, which can be measured by CVaR. In order to optimize the risk-sensitive optimization objective, we propose CPPO and provide theoretical analysis of its robustness. Moreover, we evaluate our algorithms in various MuJoCo tasks and show that CPPO obtains better performance as well as stronger robustness.

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

# A. Proofs of Theorems

In this part, we will provide the proofs of theorems proposed in paper.

## A.1. Deforming CVaR for Low Value Part

In this part, we propose the theorem 5 as below to deform VaR and CVaR for evaluating the low value part of the variable:

**Theorem 5** *For a given policy $\pi_\theta$ and its cumulative reward $D(\pi_\theta)$, we have:*

$$-\text{VaR}_\alpha(-D(\pi_\theta)) = \max\{z|F_{D(\pi_\theta)}(z) \leq 1 - \alpha\}$$

$$\begin{aligned} &- \text{CVaR}_\alpha(-D(\pi_\theta)) \\ =&E_{w\sim D(\pi_\theta)}\{w|w \leq -\text{VaR}_\alpha(-D(\pi_\theta))\}. \end{aligned}$$

*And we have:*

$$\lim_{\alpha\to1^-} -\text{CVaR}_\alpha(-D(\pi_\theta)) = \min(D(\pi_\theta)), \quad (17)$$

*and if we assume that $-\text{CVaR}_\alpha(-D(\pi_\theta)) \geq \beta$, then we have:*

$$P(D(\pi_\theta) \leq \beta) \leq 1 - \alpha.$$

Now, we will prove Theorem 5. By definition of VaR and CVaR, we have:

$$\begin{aligned} -\text{VaR}_\alpha(-Z) &= -\min\{z|F_{-Z}(z) \geq \alpha\} \\ &= -\min\{z|1 - F_Z(-z) \geq \alpha\} \\ &= \max\{-z|1 - F_Z(-z) \geq \alpha\} \\ &= \max\{z|F_Z(z) \leq 1 - \alpha\}, \end{aligned}$$

$$\begin{aligned} -\text{CVaR}_\alpha(-Z) &= -E_{w\sim Z}\{-w| - w \geq \text{VaR}_\alpha(-Z)\} \\ &= E_{w\sim Z}\{w| - w \geq \text{VaR}_\alpha(-Z)\} \\ &= E_{w\sim Z}\{w|w \leq -\text{VaR}_\alpha(-Z)\}. \end{aligned}$$

If we assume that $-\text{CVaR}_\alpha(-Z) \geq \beta$, then we have:

$$\begin{aligned} P(Z \leq \beta) &\leq P(Z \leq -\text{CVaR}_\alpha(-Z)) \\ &= P(Z \leq E_{w\sim Z}\{w|w \leq -\text{VaR}_\alpha(-Z)\}) \\ &= P(Z \leq -\text{VaR}_\alpha(-Z)) \\ &= P(Z \leq \max\{z|F_Z(z) \leq 1 - \alpha\}) \\ &= 1 - \alpha. \end{aligned}$$

So we have proven it. $\square$

## A.2. Some Properties of $\pi_c(\alpha, \beta)$

Since $\pi_c(\alpha, \beta)$ is the optimal solution of (7) and satisfies the constraint, by Theorem 5 we natural have:

$$P(D(\pi_c(\alpha, \beta)) \leq \beta) \leq 1 - \alpha,$$

which means that we can guarantee the transition probability to a catastrophic state is below a desired threshold by setting the hyper-parameters $\alpha, \beta$.

Compared with the best policy $\pi_s$ of the standard RL optimization problem (1), $\pi_c(\alpha, \beta)$ is the policy that maximizes the expected total reward in a restricted region related to hyper-parameters $\alpha, \beta$. Obviously we have $J(\pi_c(\alpha, \beta)) \leq J(\pi_s)$. However, we can also give a lower bound of $J(\pi_c(\alpha, \beta))$ as below:

**Theorem 6** *Assume there exists a constant $M > 0$ and every trajectory $\tau = (S_0, A_0, R_1, S_1, A_1, R_2, ...)$ satisfies:*

$$\sum_{t=1}^{\infty} \gamma^t R_t \leq M.$$

*We have:*

$$J(\pi_c(\alpha, \beta)) \geq \frac{J(\pi_s) - \alpha M}{1 - \alpha}.$$

Here, we will prove this Theorem. Since we assume $M$ is the upper bound of the total reward of every trajectory, we have $J(\pi_s) \leq M$. We consider two scenarios.

In the first case, if $\pi_s$ satisfies that $-CVaR_\alpha(-D(\pi_s)) \geq \beta$. Obviously, we have $\pi_c(\alpha, \beta) = \pi_s$, thus

$$J(\pi_c(\alpha, \beta)) = J(\pi_s) \geq \frac{J(\pi_s) - \alpha M}{1 - \alpha}.$$

Otherwise, we assume that $-\text{CVaR}_\alpha(-D(\pi_s)) < \beta$, Since $-\text{CVaR}_\alpha(-D(\pi_c(\alpha, \beta))) \geq \beta$, we set $B = -VaR_\alpha(-D(\pi_c(\alpha, \beta)))$ and have:

$$J(\pi_c(\alpha, \beta))$$
$$= \int_{\tau \sim \pi_c(\alpha, \beta)} p(\tau) D(\tau) d\tau$$
$$= \int_{D(\tau) \leq B} p(\tau) D(\tau) d\tau + \int_{D(\tau) > B} p(\tau) D(\tau) d\tau$$
$$\geq -\alpha CVaR(-D(\pi_c(\alpha, \beta))) + \int_{D(\tau) > B} p(\tau) B d\tau$$
$$\geq A\alpha + A(1 - \alpha)$$
$$= \beta.$$

By the similar way, we set:

$$A = -\text{VaR}_\alpha(-D(\pi_\theta)) = \max\{z | F_{D(\pi_\theta)}(z) \leq 1 - \alpha\},$$

thus

$$J(\pi_s) = \int_{\tau \sim \pi_s} p(\tau) D(\tau) d\tau$$
$$= \int_{D(\tau) \leq A} p(\tau) D(\tau) d\tau + \int_{D(\tau) > A} p(\tau) D(\tau) d\tau$$
$$= \int_{D(\tau) \leq A} p(\tau) A d\tau + \int_{D(\tau) > A} p(\tau) M d\tau$$
$$= A(1 - \alpha) + M\alpha$$
$$< \beta(1 - \alpha) + M\alpha$$
$$\leq J(\pi_c(\alpha, \beta))(1 - \alpha) + M\alpha.$$

So we have proven $J(\pi_c(\alpha, \beta)) \geq \frac{J(\pi_s) - \alpha M}{1 - \alpha}$. $\square$

### A.3. The Proof of Theorem 1

In this part, we will prove Theorem 1. We have:

$$\max_\theta J(\pi_\theta) \quad s.t. - \text{CVaR}_\alpha(-D(\pi_\theta)) \geq \beta$$
$$\Leftrightarrow \min_\theta -J(\pi_\theta) \quad s.t. \text{CVaR}_\alpha(-D(\pi_\theta)) \leq -\beta$$
$$\overset{1}{\Leftrightarrow} \min_\theta -J(\pi_\theta) \quad s.t. \min_{\nu \in R}\{\nu + \frac{1}{1 - \alpha}E[(-D(\pi_\theta) - \nu)^+]\} \leq -\beta$$
$$\Leftrightarrow \min_\theta -J(\pi_\theta) \quad s.t. \min_{\nu \in R}\{-\nu + \frac{1}{1 - \alpha}E[(-D(\pi_\theta) + \nu)^+]\} \leq -\beta$$
$$\Leftrightarrow \min_{\theta, \nu} -J(\pi_\theta) \quad s.t. -\nu + \frac{1}{1 - \alpha}E[(-D(\pi_\theta) + \nu)^+] \leq -\beta.$$

Here we derive a formula 1 since CVaR owns the property (Chow et al., 2015b):

$$\text{CVaR}_\alpha(Z) = \min_{\eta \in R}\left\{\eta + \frac{1}{1 - \alpha}E[(Z - \eta)^+]\right\}. \quad (18)$$

So we have proven it. $\square$

### A.4. The Proof of Theorem 2

In this part, we will prove Theorem 2.

We will calculate the gradient $\partial_\nu L(\theta, \nu, \lambda), \bigtriangledown_\theta L(\theta, \nu, \lambda)$ and $\bigtriangledown_\lambda L(\theta, \nu, \lambda)$ of the function $L(\theta, \nu, \lambda)$ by using the methods in (Chow & Ghavamzadeh, 2014a):

$$L(\theta, \nu, \lambda) = -J(\pi_\theta) + \lambda(-\nu + \frac{1}{1 - \alpha}E[(-D(\pi_\theta) + \nu)^+] + \beta).$$

First we can expand the expectation:

$$L(\theta, \nu, \lambda)$$
$$= -J(\pi_\theta) + \lambda(-\nu + \frac{1}{1-\alpha}E[(-D(\pi_\theta) + \nu)^+] + \beta)$$
$$= -\sum_\xi P_\theta(\xi)D(\xi) - \lambda\nu$$
$$+ \frac{\lambda}{1-\alpha}\sum_\xi P_\theta(\xi)(-D(\xi) + \nu)^+ + \lambda\beta.$$

We can see that $P_\theta(\xi)$ will only depend on $\theta$ and $\xi$, so we have easily calculate the gradient of $\lambda$:

$$\nabla_\lambda L(\theta, \nu, \lambda)$$
$$= -\nu + \frac{1}{1-\alpha}\sum_\xi P_\theta(\xi)(-D(\xi) + \nu)^+ + \beta$$
$$= -\nu + \frac{1}{1-\alpha}E_{\xi\sim\pi_\theta}(-D(\xi) + \nu)^+ + \beta.$$

Then we calculate the gradient of $\nu$. Since $(D(\xi) - \nu)^+$ isn't differentiable to $\nu$ at the point of $\nu = D(\xi)$, so we consider its semi gradient:

$$\partial_\nu(-D(\xi) + \nu)^+ = \begin{cases} 0 & \nu < D(\xi) \\ q(0 \le q \le 1) & \nu = D(\xi) \\ 1 & \nu > D(\xi) \end{cases}$$

And we can calculate the gradient of $\nu$ as below:

$$\partial_\nu L(\theta, \nu, \lambda)$$
$$= -\lambda + \frac{\lambda}{1-\alpha}\sum_\xi P_\theta(\xi)\partial_\nu(-D(\xi) + \nu)^+$$
$$= -\lambda + \frac{\lambda}{1-\alpha}\sum_\xi P_\theta(\xi)\mathbf{1}\{\nu > D(\xi)\}$$
$$+ \frac{\lambda q}{1-\alpha}\sum_\xi P_\theta(\xi)\mathbf{1}\{\nu = D(\xi)\}$$
$$= -\lambda + \frac{\lambda}{1-\alpha}\sum_\xi P_\theta(\xi)\mathbf{1}\{\nu \ge D(\xi)\} \quad (let\ q = 1)$$
$$= -\lambda + \frac{\lambda}{1-\alpha}E_{\xi\sim\pi_\theta}\mathbf{1}\{\nu \ge D(\xi)\}).$$

Finally, we will calculate the gradient of $\theta$.

$$\nabla_\theta L(\theta, \nu, \lambda)$$
$$= -\sum_\xi \nabla_\theta P_\theta(\xi)D(\xi) + \frac{\lambda}{1-\alpha}\sum_\xi \nabla_\theta P_\theta(\xi)(-D(\xi) + \nu)^+$$
$$= \sum_\xi \nabla_\theta P_\theta(\xi)(-D(\xi) + \frac{\lambda}{1-\alpha}(-D(\xi) + \nu)\mathbf{1}\{\nu \ge D(\xi)\})$$
$$= \sum_\xi (\nabla_\theta \log P_\theta(\xi))P_\theta(\xi)$$
$$(-D(\xi) + \frac{\lambda}{1-\alpha}(-D(\xi) + \nu)\mathbf{1}\{\nu \ge D(\xi)\})$$
$$= -\sum_\xi (\nabla_\theta \log P_\theta(\xi))P_\theta(\xi)D(\xi)$$
$$+ \sum_\xi (\nabla_\theta \log P_\theta(\xi))P_\theta(\xi)\frac{\lambda(\nu - D(\xi))}{1-\alpha}\mathbf{1}\{\nu \ge D(\xi)\}$$
$$= -E_{\xi\sim\pi_\theta}(\nabla_\theta \log P_\theta(\xi))\left(D(\xi) - \frac{\lambda}{1-\alpha}(-D(\xi) + \nu)^+\right).$$

So we have calculated these three gradient and prove Theorem 2. $\square$

### A.5. The Proof of Theorem 3 and Theorem 4

Before proving Theorem 3 and Theorem 4, we first examine a property of $d_\mathcal{M}^\pi$:

**Lemma 1** *For any state $s \in \mathcal{S}$, we have:*

$$d_\mathcal{M}^\pi(s) = (1-\gamma)P(s_0 = s) + \gamma\sum_{s'} d_\mathcal{M}^\pi(s')\sum_a \pi(a|s)P(s'|s, a). \tag{19}$$

Here we'll prove this lemma. By the definition of $d_\mathcal{M}^\pi(s)$, we have:

$$d_\mathcal{M}^\pi(s) - (1 - \gamma)P(s_0 = s)$$
$$= (1 - \gamma)\sum_{t=1}^\infty \sum_{s'} \gamma^t P(s_{t-1} = s', s_t = s|\pi, \mathcal{M})$$
$$= (1 - \gamma)\sum_{t=0}^\infty \sum_{s'} \gamma^{t+1} P(s_t = s'|\pi, \mathcal{M})P(s_{t+1} = s|s_t = s', \pi, \mathcal{M})$$
$$= \gamma\sum_{s'}\left[(1 - \gamma)\sum_{t=0}^\infty \gamma^t P(s_t = s'|\pi, \mathcal{M})\right]P(s_1 = s|s_0 = s', \pi, \mathcal{M})$$
$$= \gamma\sum_{s'} d_\mathcal{M}^\pi(s')P(s_1 = s|s_0 = s', \pi, \mathcal{M})$$
$$= \gamma\sum_{s'} d_\mathcal{M}^\pi(s')\sum_a \pi(a|s)P(s'|s, a). \tag{20}$$

Thus we have proven it. □

Now we will prove Theorem 3. Considering the bellman equation of value function of $\pi, \hat{\pi}_\nu$ in $\mathcal{M}$, we have:

$$
V_{\mathcal{M},\pi}(s)
= \sum_a \pi(a|s)[R(s,a) + \gamma \sum_{s'} P(s'|s,a)V_{\mathcal{M},\pi}(s')],
$$
$$
V_{\mathcal{M},\hat{\pi}_\nu}(s)
= \sum_a \pi(a|\nu(s))[R(s,a) + \gamma \sum_{s'} P(s'|s,a)V_{\mathcal{M},\hat{\pi}_\nu}(s')].
$$

By subtracting two value functions, we can deduce:

$$
V_{\mathcal{M},\hat{\pi}_\nu}(s) - V_{\mathcal{M},\pi}(s)
$$
$$
= \gamma \sum_a (\pi(a|\nu(s)) - \pi(a|s)) \sum_{s'} P(s'|s,a)V_{\mathcal{M},\pi}(s')
$$
$$
+ \gamma \sum_a \pi(a|\nu(s)) \sum_{s'} P(s'|s,a)(V_{\mathcal{M},\hat{\pi}_\nu}(s') - V_{\mathcal{M},\pi}(s'))
$$
$$
+ \sum_a [\pi(a|\nu(s)) - \pi(a|s)]R(s,a).
$$
$$(21)$$

Since equation (21) satisfies for every state $s$, thus we calculate the expectation of equation (21) for $s \sim d_{\mathcal{M}}^{\hat{\pi}_\nu}$:

$$
\sum_s d_{\mathcal{M}}^{\hat{\pi}_\nu}(s)[V_{\mathcal{M},\hat{\pi}_\nu}(s) - V_{\mathcal{M},\pi}(s)]
$$
$$
= \gamma \sum_s d_{\mathcal{M}}^{\hat{\pi}_\nu}(s) \sum_a (\pi(a|\nu(s)) - \pi(a|s))
$$
$$
\sum_{s'} P(s'|s,a)V_{\mathcal{M},\pi}(s')
$$
$$
+ \gamma \sum_s d_{\mathcal{M}}^{\hat{\pi}_\nu}(s) \sum_a \pi(a|\nu(s))
$$
$$
\sum_{s'} P(s'|s,a)(V_{\mathcal{M},\hat{\pi}_\nu}(s') - V_{\mathcal{M},\pi}(s'))
$$
$$
+ \sum_s d_{\mathcal{M}}^{\hat{\pi}_\nu}(s) \sum_a [\pi(a|\nu(s)) - \pi(a|s)]R(s,a)
$$
$$
= \gamma \sum_s d_{\mathcal{M}}^{\hat{\pi}_\nu}(s) \sum_a (\pi(a|\nu(s)) - \pi(a|s))
$$
$$
\sum_{s'} P(s'|s,a)V_{\mathcal{M},\pi}(s')
$$
$$
+ \sum_{s'} (V_{\mathcal{M},\hat{\pi}_\nu}(s') - V_{\mathcal{M},\pi}(s'))
$$
$$
\left[ \gamma \sum_s d_{\mathcal{M}}^{\hat{\pi}_\nu}(s) \sum_a \pi(a|\nu(s))P(s'|s,a) \right]
$$
$$
+ \sum_s d_{\mathcal{M}}^{\hat{\pi}_\nu}(s) \sum_a [\pi(a|\nu(s)) - \pi(a|s)]R(s,a).
$$
$$(22)$$

By Lemma 1, we have:

$$
\sum_s d_{\mathcal{M}}^{\hat{\pi}_\nu}(s)[V_{\mathcal{M},\hat{\pi}_\nu}(s) - V_{\mathcal{M},\pi}(s)]
$$
$$
= \gamma \sum_s d_{\mathcal{M}}^{\hat{\pi}_\nu}(s) \sum_a (\pi(a|\nu(s)) - \pi(a|s))
$$
$$
\sum_{s'} P(s'|s,a)V_{\mathcal{M},\pi}(s')
$$
$$
+ \sum_{s'} (V_{\mathcal{M},\hat{\pi}_\nu}(s') - V_{\mathcal{M},\pi}(s'))
$$
$$
\left[ d_{\mathcal{M}}^{\hat{\pi}_\nu}(s') - (1-\gamma)P(s_0 = s') \right]
$$
$$
+ \sum_s d_{\mathcal{M}}^{\hat{\pi}_\nu}(s) \sum_a [\pi(a|\nu(s)) - \pi(a|s)]R(s,a).
$$
$$(23)$$

By moving the second term of the right part in (23) to the left part, we can deduce:

$$
(1-\gamma) \sum_{s'} (V_{\mathcal{M},\hat{\pi}_\nu}(s') - V_{\mathcal{M},\pi}(s'))P(s_0 = s')
$$
$$
= \gamma \sum_s d_{\mathcal{M}}^{\hat{\pi}_\nu}(s) \sum_a (\pi(a|\nu(s)) - \pi(a|s))
$$
$$
\sum_{s'} P(s'|s,a)V_{\mathcal{M},\pi}(s')
$$
$$
+ \sum_s d_{\mathcal{M}}^{\hat{\pi}_\nu}(s) \sum_a [\pi(a|\nu(s)) - \pi(a|s)]R(s,a),
$$
$$(24)$$

thus:

$$
(1-\gamma)(J_{\mathcal{M}}(\hat{\pi}_\nu) - J_{\mathcal{M}}(\pi))
$$
$$
= (1-\gamma) \sum_{s'} (V_{\mathcal{M},\hat{\pi}_\nu}(s') - V_{\mathcal{M},\pi}(s'))P(s_0 = s')
$$
$$
= \gamma \sum_s d_{\mathcal{M}}^{\hat{\pi}_\nu}(s) \sum_a (\pi(a|\nu(s)) - \pi(a|s)) \sum_{s'} P(s'|s,a)V_{\mathcal{M},\pi}(s')
$$
$$
+ \sum_s d_{\mathcal{M}}^{\hat{\pi}_\nu}(s) \sum_a [\pi(a|\nu(s)) - \pi(a|s)]R(s,a)
$$
$$
= \gamma E_{s \sim d_{\mathcal{M}}^{\hat{\pi}_\nu}} E_{a \sim \pi(\cdot|\nu(s))} \left( 1 - \frac{\pi(a|s)}{\pi(a|\nu(s))} \right) E_{s' \sim P(\cdot|s,a)} V_{\mathcal{M},\pi}(s')
$$
$$
+ E_{s \sim d_{\mathcal{M}}^{\hat{\pi}_\nu}} E_{a \sim \pi(\cdot|\nu(s))} \left( 1 - \frac{\pi(a|s)}{\pi(a|\nu(s))} \right) R(s,a).
$$

And we can prove:

$$
J_{\mathcal{M}}(\pi) - J_{\mathcal{M}}(\hat{\pi}_\nu)
$$
$$
= \frac{\gamma}{1-\gamma} E_{s \sim d_{\mathcal{M}}^{\hat{\pi}_\nu}} E_{a \sim \pi(\cdot|\nu(s))} \left( 1 - \frac{\pi(a|s)}{\pi(a|\nu(s))} \right) E_{s' \sim P(\cdot|s,a)} V_{\mathcal{M},\pi}(s')
$$
$$
+ \frac{1}{1-\gamma} E_{s \sim d_{\mathcal{M}}^{\hat{\pi}_\nu}} E_{a \sim \pi(\cdot|\nu(s))} \left( 1 - \frac{\pi(a|s)}{\pi(a|\nu(s))} \right) R(s,a).
$$
$$(25)$$

Since $E_{a \sim \pi(\cdot|\nu(s))} \left(1 - \frac{\pi(a|s)}{\pi(a|\nu(s))}\right) = 0$, we can subtract a benchmark, which will not affect its value. Specially, we set $b = \max_{s'} V_{\mathcal{M},\pi}(s') - \min_{s'} V_{\mathcal{M},\pi}(s')$ and we have $|V_{\mathcal{M},\pi}(s) - b| \le \frac{b}{2}$ for every state $s$, thus we can prove:

$$
\begin{aligned}
&|J_{\mathcal{M}}(\pi) - J_{\mathcal{M}}(\hat{\pi}_\nu)| \\
\le& \frac{\gamma}{1-\gamma} E_{s \sim d_{\mathcal{M}}^{\hat{\pi}_\nu}} E_{a \sim \pi(\cdot|\nu(s))} \left|1 - \frac{\pi(a|s)}{\pi(a|\nu(s))}\right| \\
&\left|E_{s' \sim P(\cdot|s,a)} V_{\mathcal{M},\pi}(s') - b\right| \\
+& \frac{1}{1-\gamma} E_{s \sim d_{\mathcal{M}}^{\hat{\pi}_\nu}} E_{a \sim \pi(\cdot|\nu(s))} \left|1 - \frac{\pi(a|s)}{\pi(a|\nu(s))}\right| |R(s,a)| \\
\le& \frac{\gamma}{1-\gamma} E_{s \sim d_{\mathcal{M}}^{\hat{\pi}_\nu}} E_{a \sim \pi(\cdot|\nu(s))} \left|1 - \frac{\pi(a|s)}{\pi(a|\nu(s))}\right| \frac{b}{2} \\
+& \frac{1}{1-\gamma} E_{s \sim d_{\mathcal{M}}^{\hat{\pi}_\nu}} E_{a \sim \pi(\cdot|\nu(s))} \left|1 - \frac{\pi(a|s)}{\pi(a|\nu(s))}\right| \max_{s,a} |R(s,a)| \\
\le& \frac{\gamma}{1-\gamma} E_{s \sim d_{\mathcal{M}}^{\hat{\pi}_\nu}} \sum_a |\pi(a|\nu(s)) - \pi(a|s)| \frac{b}{2} \\
+& \frac{1}{1-\gamma} E_{s \sim d_{\mathcal{M}}^{\hat{\pi}_\nu}} \sum_a |\pi(a|\nu(s)) - \pi(a|s)| \max_{s,a} |R(s,a)| \\
=& \frac{\gamma}{1-\gamma} E_{s \sim d_{\mathcal{M}}^{\hat{\pi}_\nu}} \max_s D_{TV}(\pi(\cdot|s), \pi(\cdot|\nu(s))) b \\
+& \frac{2}{1-\gamma} E_{s \sim d_{\mathcal{M}}^{\hat{\pi}_\nu}} \max_s D_{TV}(\pi(\cdot|s), \pi(\cdot|\nu(s)) \max_{s,a} |R(s,a)|.
\end{aligned}
\tag{26}
$$

Finally, we will prove that our bound is tighter than the bound in (Zhang et al., 2020):

$$
\begin{aligned}
&|J_{\mathcal{M}}(\pi) - J_{\mathcal{M}}(\hat{\pi}_\nu)| \\
\le& \frac{\gamma}{1-\gamma} \max_s D_{TV}(\pi(\cdot|s), \pi(\cdot|\nu(s))) b \\
+& \frac{2}{1-\gamma} \max_s D_{TV}(\pi(\cdot|s), \pi(\cdot|\nu(s)) \max_{s,a} |R(s,a)| \\
\le& \frac{2\gamma}{1-\gamma} \max_s D_{TV}(\pi(\cdot|s), \pi(\cdot|\nu(s))) \max_s |V_{\mathcal{M},\pi}(s)| \\
+& \frac{2}{1-\gamma} \max_s D_{TV}(\pi(\cdot|s), \pi(\cdot|\nu(s)) \max_{s,a} |R(s,a)| \\
\le& \left(\frac{2\gamma}{(1-\gamma)^2} + \frac{2}{1-\gamma}\right) \max_s D_{TV}(\pi(\cdot|s), \pi(\cdot|\nu(s))) \\
& \max_{s,a} |R(s,a)|.
\end{aligned}
\tag{27}
$$

Thus we have proven it. $\quad\square$

Finally, we will prove Theorem 4 by using the similar method of Theorem 3. Similarly, considering the bellman equation of value function of $\pi$ in $\mathcal{M}, \hat{\mathcal{M}}$, we have:

$$
\begin{aligned}
V_{\mathcal{M},\pi}(s) &= \sum_a \pi(a|s)[R(s,a) + \gamma \sum_{s'} P(s'|s,a)V_{\mathcal{M},\pi}(s')], \\
V_{\hat{\mathcal{M}},\pi}(s) &= \sum_a \pi(a|s)[R(s,a) + \gamma \sum_{s'} \hat{P}(s'|s,a)V_{\hat{\mathcal{M}},\pi}(s')].
\end{aligned}
\tag{28}
$$

By subtracting them, we have:

$$
\begin{aligned}
&V_{\hat{\mathcal{M}},\pi}(s) - V_{\mathcal{M},\pi}(s) \\
=& \gamma \sum_a \pi(a|s) \sum_{s'} (\hat{P}(s'|s,a) - P(s'|s,a)) V_{\mathcal{M},\pi}(s') \\
+& \gamma \sum_a \pi(a|s) \sum_{s'} \hat{P}(s'|s,a)(V_{\hat{\mathcal{M}},\pi}(s') - V_{\mathcal{M},\pi}(s')).
\end{aligned}
\tag{29}
$$

Since equation (21) satisfies for every state $s$, thus we calculate the expectation of equation (21) for $s \sim d_{\mathcal{M}}^{\hat{\pi}_\nu}$ and use Lemma 1:

$$
\begin{aligned}
&\sum_s d_{\hat{\mathcal{M}}}^\pi(s)[V_{\hat{\mathcal{M}},\pi}(s) - V_{\mathcal{M},\pi}(s)] \\
=& \gamma \sum_s d_{\hat{\mathcal{M}}}^\pi(s) \sum_a \pi(a|s) \sum_{s'} (\hat{P}(s'|s,a) - P(s'|s,a)) V_{\mathcal{M},\pi}(s') \\
+& \gamma \sum_s d_{\hat{\mathcal{M}}}^\pi(s) \sum_a \pi(a|s) \sum_{s'} \hat{P}(s'|s,a)(V_{\hat{\mathcal{M}},\pi}(s') - V_{\mathcal{M},\pi}(s')) \\
=& \gamma \sum_s d_{\hat{\mathcal{M}}}^\pi(s) \sum_a \pi(a|s) \sum_{s'} (\hat{P}(s'|s,a) - P(s'|s,a)) V_{\mathcal{M},\pi}(s') \\
+& \sum_{s'} (V_{\hat{\mathcal{M}},\pi}(s') - V_{\mathcal{M},\pi}(s')) \left[\gamma \sum_s d_{\hat{\mathcal{M}}}^\pi(s) \sum_a \pi(a|s)\hat{P}(s'|s,a)\right] \\
=& \gamma \sum_s d_{\hat{\mathcal{M}}}^\pi(s) \sum_a \pi(a|s) \sum_{s'} (\hat{P}(s'|s,a) - P(s'|s,a)) V_{\mathcal{M},\pi}(s') \\
+& \sum_{s'} (V_{\hat{\mathcal{M}},\pi}(s') - V_{\mathcal{M},\pi}(s')) \left[d_{\hat{\mathcal{M}}}^\pi(s') - (1-\gamma)P(s_0 = s')\right].
\end{aligned}
\tag{30}
$$

Similarly, by moving the second term of the right part in (30) to the left part, we can deduce:

$$
\begin{aligned}
&(1-\gamma) \sum_{s'} (V_{\hat{\mathcal{M}},\pi}(s') - V_{\mathcal{M},\pi}(s'))P(s_0 = s') \\
=& \gamma \sum_s d_{\hat{\mathcal{M}}}^\pi(s) \sum_a \pi(a|s) \sum_{s'} (\hat{P}(s'|s,a) - P(s'|s,a)) V_{\mathcal{M},\pi}(s'),
\end{aligned}
\tag{31}
$$

thus:

$$(1-\gamma)(J_{\hat{\mathcal{M}}}(\pi) - J_{\mathcal{M}}(\pi))$$

$$=(1-\gamma)\sum_{s'}(V_{\hat{\mathcal{M}},\pi}(s') - V_{\mathcal{M},\pi}(s'))P(s_0 = s')$$

$$=\gamma \sum_s d_{\hat{\mathcal{M}}}^{\pi}(s) \sum_a \pi(a|s)$$

$$\sum_{s'}(\hat{P}(s'|s,a) - P(s'|s,a))V_{\mathcal{M},\pi}(s') \qquad (32)$$

$$=\gamma E_{s\sim d_{\hat{\mathcal{M}}}^{\pi}} E_{a\sim\pi(\cdot|s)}$$

$$E_{s'\sim\hat{P}(\cdot|s,a)}\left(1 - \frac{P(s'|s,a)}{\hat{P}(s'|s,a)}\right)V_{\mathcal{M},\pi}(s').$$

Thus we have proven:

$$J_{\mathcal{M}}(\pi) - J_{\hat{\mathcal{M}}}(\pi)$$

$$=\frac{\gamma}{1-\gamma} E_{s\sim d_{\hat{\mathcal{M}}}^{\pi}} E_{a\sim\pi(\cdot|s)} \qquad (33)$$

$$E_{s'\sim\hat{P}(\cdot|s,a)}\left(1 - \frac{P(s'|s,a)}{\hat{P}(s'|s,a)}\right)V_{\mathcal{M},\pi}(s').$$

Similarly, we set $b = \max_{s'} V_{\mathcal{M},\pi}(s') - \min_{s'} V_{\mathcal{M},\pi}(s')$ and we have $|V_{\mathcal{M},\pi}(s) - b| \le \frac{b}{2}$ for every state $s$, thus we can prove:

$$|J_{\mathcal{M}}(\pi) - J_{\mathcal{M}}(\hat{\pi}_{\nu})|$$

$$\le \frac{\gamma}{1-\gamma} E_{s\sim d_{\hat{\mathcal{M}}}^{\pi}} E_{a\sim\pi(\cdot|s)} E_{s'\sim\hat{P}(\cdot|s,a)} \left|1 - \frac{P(s'|s,a)}{\hat{P}(s'|s,a)}\right|$$

$$|V_{\mathcal{M},\pi}(s') - b|$$

$$\le \frac{\gamma}{1-\gamma} E_{s\sim d_{\hat{\mathcal{M}}}^{\pi}} E_{a\sim\pi(\cdot|s)} E_{s'\sim\hat{P}(\cdot|s,a)} \left|1 - \frac{P(s'|s,a)}{\hat{P}(s'|s,a)}\right| \frac{b}{2}$$

$$= \frac{\gamma}{1-\gamma} E_{s\sim d_{\hat{\mathcal{M}}}^{\pi}} E_{a\sim\pi(\cdot|s)} \sum_{s'} \left|\hat{P}(s'|s,a) - P(s'|s,a)\right| \frac{b}{2}$$

$$= \frac{\gamma}{1-\gamma} E_{s\sim d_{\hat{\mathcal{M}}}^{\pi}} E_{a\sim\pi(\cdot|s)} D_{TV}(P(\cdot|s,a), \hat{P}(\cdot|s,a))b.$$

$$(34)$$

Thus we have proven it. □

## B. Pseudo code of CPPO

In this part, we will provide the pseudo code of our algorithm CPPO.

---

**Algorithm 1** CVaR Proximal Policy Optimization(CPPO)

---

**Require:** confidence level $\alpha$ and reward tolerance $\beta$

**Ensure:** $\theta$ of parameterized policy $\pi_\theta$(always be random policy), $\phi$ of parameterized value function $V_\phi$.

   **for** $k = 1, 2, ..., N_{iter}$ **do**

      Generate $N$ trajectories $\mathcal{D}_k = \{\xi_i\}_{i=1}^N$ by following the current policy $\pi_\theta$.

      Compute reward $\hat{R}_i^t$ of each state $s_{i,t}$ in each trajectory $\xi_i$ and the cumulative reward $D(\xi_i)$.

      Compute advantage estimates $\hat{A}_i^t$ of each state $s_{i,t}$ in each trajectory $\xi_i$.

      Update parameters respectively:

$$\eta \leftarrow \eta - lr_\eta \left(-\lambda + \frac{\lambda}{N(1-\alpha)} \sum_{i=1}^N \mathbf{1}\{\eta \ge D(\xi_i)\}\right)$$

$$\theta \leftarrow \theta + lr_\theta \frac{1}{NT} \sum_{i=1}^N \sum_{t=0}^T \nabla_\theta \min\left(\frac{\pi_\theta(a_i^t|s_i^t)}{\pi_{\theta_k}(a_i^t|s_i^t)} \hat{A}_i^t, g(\epsilon, \hat{A}_i^t)\right)$$

$$- lr_\theta \frac{1}{N} \sum_{i=1}^N (\nabla_\theta \log P_\theta(\xi_i)) \frac{\lambda}{1-\alpha}(-D(\xi_i) + \eta)$$

$$\mathbf{1}\{\eta \ge D(\xi_i)\}$$

$$\lambda \leftarrow \lambda + lr_\lambda \left(-\eta + \frac{\sum_{i=1}^N(-D(\xi_i) + \eta)^+}{N(1-\alpha)} + \beta\right)$$

$$\phi \leftarrow \phi + lr_\phi \left(\frac{1}{NT} \sum_{i=1}^N \sum_{t=0}^T 2(V_\phi(s_{i,t}) - \hat{R}_i^t)\nabla_\phi V_\phi(s_{i,t})\right)$$

   **end for**

---

# C. Details of Experiments

In this part, we will provide the details of our experiments.

## C.1. Performance in Training Stage

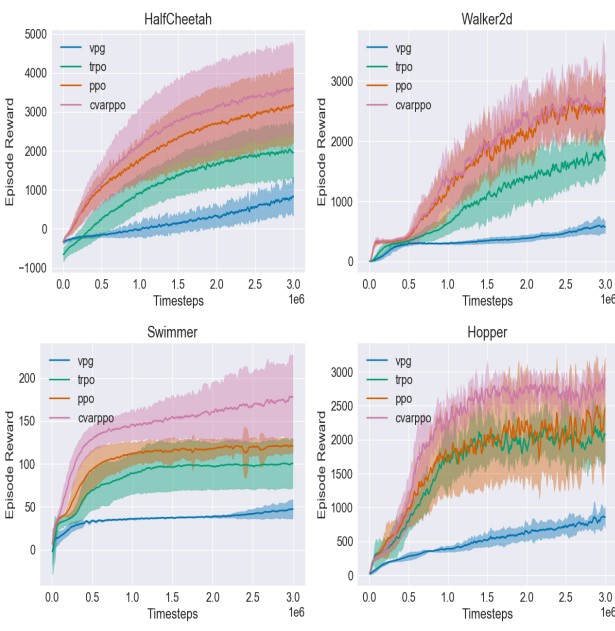

Figure 1. The abscissa is the number of steps interacting with the environment, and the ordinate is the performance of the agent.

Firstly, in order to compare the performance of VPG, TRPO, PPO and CPPO in the training stage, we train 10 policies with different random seed for each algorithm, and plot the mean and variance of the ten policies as a function of the training step as shown in Figure 1. The four subgraphs in Figure 1 represent the experimental results on halfcheetah, walker2d, swimmer and hopper respectively. The abscissa represents the number of timesteps for the agent to interact with the environment, and the ordinate represents the cumulative expected reward of the agent. The solid line represents the average reward of 10 strategies, and the dotted line represents the variance of them. Among them, blue represents VPG, green represents TRPO, orange represents PPO, and pink represents our CPPO. As shown in Figure 1, CPPO has achieved significant performance improvement on HalfCheetah, Swimmer and Hopper. Table 1 shows the results of mean plus and minus variance trained by each algorithm.

## C.2. Robustness under State Observation Disturbance in Test Stage

Next, we consider the robustness and security of the trained model against state observation disturbance. We test the performance of the trained model under the disturbance of observation state and draw the Figure 2, where the abscissa

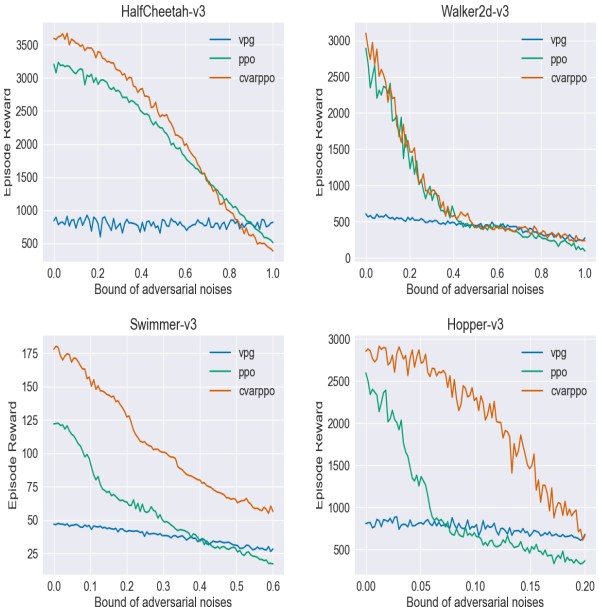

Figure 2. The abscissa is the range of the disturbance, and the ordinate is the average performance of the algorithm under the state disturbance.

represents the size of the disturbance and the ordinate represents the average reward under the disturbance. As shown in the figure, CPPO has made significant progress in Swimmer and Hopper.

## C.3. Robustness under Transition Probability Disturbance in Test Stage

Finally, we compare the performance of different algorithms under the disturbance of transition probability. It is noted that MuJoCo is a simulator modeled on the physical world, thus we can modify the transition probability of the environment by modifying the environment parameters. We mainly choose to modify the mass of the robot to achieve the purpose of modifying the transition probability. The default mass of the environments HalfCheetah-v3, Walker2d-v3, Swimmer-v3 and Hopper-v3 are 6.36, 3.53, 34.6 and 3.53, We consider the case of larger mass and smaller mass respectively. Therefore, we draw Figure 3, where the abscissa represents the mass of the agent and the ordinate represents the performance of the strategy under different mass conditions. From Figure 3, we can see that the performance of all algorithms decreases to a certain extent with the change of agent quality (whether it becomes larger or smaller), and the degree of decline is positively correlated with the quality change, which is consistent with our theoretical analysis, that is, the upper bound of the performance difference of the algorithm is related to the size of the transition probability disturbance. At the same time, we can see that CPPO has achieved better robustness improvement in different tasks.

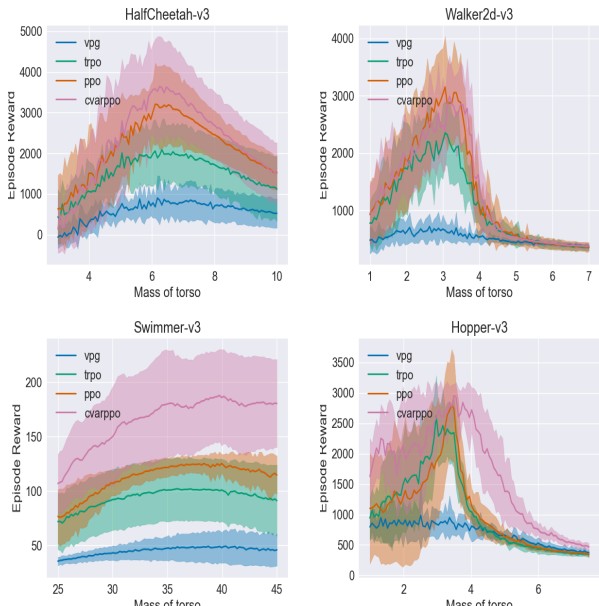

*Figure 3.* The abscissa is the mass of the agent, and the ordinate is the average performance of the algorithm when the mass changes.