# OpenReview forum: "Towards Safe Reinforcement Learning via Constraining Conditional Value at Risk"
_ICML.cc/2021/Workshop/AML — ICML 2021 Workshop AML Poster_

### Official Review · Reviewer_nqPf · 2021-06-21
**An excellent work in theoretical aspects**

**Rating:** Accept
**Confidence:** 5

**Review:**

This paper proposed a method combined RL algorithms with CVaR. The empirical results outperforms other baselines in MuJoCo. Besides, this paper is well organized and the theoretical analysis is insightful.

less crucial problems:
1. Compared to the main part of the paper, the appendix is not well-written. For example, the title in the appendix is inconsistent snd the text in Fig.2 is unclear.
2. The results in the appendix appear to weaken the persuasiveness of the conclusions.

---

### Decision · Program_Chairs · 2021-06-21

**Decision:**

Accept (Poster)

**Comment:**

A good work to combine RL with CVaR. The authors can further improve the paper by addressing the reviewer's comments.